# Tracking County-level Cooking Emissions and Their Drivers in China from 1990 to 2021 by Ensemble Machine Learning

Zeqi Li[1,2], Bin Zhao[1,2], Shengyue Li[1,2], Zhezhe Shi[1,2], Dejia Yin[1,2], Qingru Wu[1,2], Fenfen Zhang[1,2,3], Xiao Yun[4,5], Guanghan Huang[6], Yun Zhu[7], Shuxiao Wang[1,2]

[1]State Key Joint Laboratory of Environmental Simulation and Pollution Control, School of Environment, Tsinghua University, Beijing, 100084, China
[2]State Environmental Protection Key Laboratory of Sources and Control of Air Pollution Complex, Beijing 100084, China
[3]Department of Environment, Yangtze Delta Region Institute of Tsinghua University, Zhejiang, Jiaxing 314006, China
[4]China Energy Longyuan Environmental Protection Co., Ltd., Beijing 100039, China
[5]National Engineering Research Center of New Energy Power Generation, North China Electric Power University, Beijing 102206, China
[6]Beijing Municipal Research Institute of Eco-Environmental Protection, Beijing 100037, China
[7]Guangdong Provincial Key Laboratory of Atmospheric Environment and Pollution Control, College of Environment and Energy, South China University of Technology, Guangzhou, 510006, China

*Correspondence to*: Shuxiao Wang (shxwang@tsinghua.edu.cn)

**Abstract.** Cooking emissions are a significant source of $PM_{2.5}$, posing considerable public health risks due to their high toxicity and proximity to densely populated areas. Despite their importance, there is currently a lack of an accurate, long-term, high-resolution national cooking emission inventory in China, primarily due to the challenges in obtaining high-quality activity level data over extended periods and at fine spatial scales. Here, we address these limitations by leveraging advanced
machine learning techniques to predict activity levels and further estimate emissions.

Specifically, we develop an ensemble model of machine learning algorithms—Random Forest (RF), eXtreme Gradient Boosting (XGBoost), Multilayer Perceptron Neural Network (MLP), and Deep Neural Networks (DNN)—to accurately predict cooking activity levels across Chinese counties based on statistical indicators related to population, economy, and the catering industry. The ensemble machine learning model demonstrates exceptional generalization and transferability
($R^2$=0.892-0.989), outperforming traditional statistical models and individual machine learning models. Unlike previous inventories that rely on simplistic proxy data such as population for calculation and downscaling, our inventory precisely calculates county-level cooking emissions, providing more accurate emission estimates and spatial distributions. Furthermore, we incorporate critical but previously missing toxic pollutants, such as ultrafine particles (UFPs) and polycyclic aromatic hydrocarbons (PAHs), into the national cooking emission inventory. Therefore, we develop China's first
county-level cooking emission inventory, spanning from 1990 to 2021, with high spatial resolution and wide pollutant coverage.

According to our inventory, in 2021, China's total cooking emissions of organics in the full volatility range, $PM_{2.5}$, UFPs, and PAHs are 997 kt, 408 kt, $6.50 \times 10^{25}$ particles, and 15.8 kt, respectively. From 1990 to 2021, emissions of these

pollutants increased by over 65%, and their spatiotemporal trends were affected to varying degrees by external factors, such as population migration, economic development, pollution control policies, and the pandemic at different periods. We further analyze the contribution patterns of key driving factors, such as urbanization rate, population, and pollution control, to emission changes. Notably, driver analysis reveals that existing control measures are insufficient to curb the rapid growth of emissions, necessitating enhanced controls. Regarding control strategies, our county-level inventory finds that 62.3% of China's organic emissions are concentrated in 30% of the counties, which are densely populated and occupy only 14.4% of the national land area. Therefore, prioritizing control of these areas will be an efficient and targeted strategy. Our research provides crucial data and insights for understanding the impact of cooking emissions on air pollution and health, aiding in policy development. Our long-term, high-resolution emission datasets are publicly available at https://doi.org/10.6084/m9.figshare.26085487 (Li et al, 2025).

## 1 Introduction

Cooking activities, through the heating and processing of oil and food ingredients, emit large amounts of pollutants, posing significant harm to air quality and human health. Cooking emissions are one of the major sources of organic aerosols (OA, the organic component of $PM_{2.5}$) in urban areas (Lee et al., 2015; Logue et al., 2014; Zhao and Zhao, 2018). Source apportionment results indicate that cooking organic aerosols account for 5%-37% of the total OA concentration in various urban atmospheres (Abdullahi et al., 2013; Huang et al., 2021; Mohr et al., 2012). Moreover, cooking emissions contain multiple hazardous components, such as ultrafine particles (UFPs) and polycyclic aromatic hydrocarbons (PAHs), which are linked to health problems including cardiovascular disease, oxidative stress, and lung cancer (Kim et al., 2024; Lin et al., 2022b; Naseri et al., 2024; Xu et al., 2020). Experiments have proved that both gaseous organics and $PM_{2.5}$ emitted from cooking exhibit much more negative biological effects like cytotoxicity compared to ambient $PM_{2.5}$ (Guo et al., 2023). Consequently, cooking emissions can increase $PM_{2.5}$ concentrations and toxicity, thereby exacerbating air pollution and associated disease burdens (Chafe et al., 2014; Wang et al., 2017; Zhang et al., 2024a). Given that cooking activities predominantly occur in densely populated areas, they pose a substantial public health risk. Therefore, the long-term high-spatial-resolution emission inventories are critical for assessing the impacts of cooking emissions on human health, as they support exposure analysis studies across different locations and periods.

Chinese cooking emissions exhibit unique characteristics that require special consideration in emission inventory development (Zhao and Zhao, 2018). The widespread use of high-temperature oils, various seasonings, and special techniques like stir-frying generates pollutants with complex chemical compositions (Chen et al., 2018; Zhao and Zhao, 2018). Furthermore, cooking styles vary greatly across different regions in China, which have historically developed into multiple distinct cuisine systems (Li et al., 2023; Lin et al., 2022b). To accurately capture these emission patterns and quantify their impact, the emission estimate needs to incorporate cuisine-specific multi-pollutant emission factors while

explicitly accounting for regional variations through spatial resolution representation (Lin et al., 2022b; Zhao and Zhao, 2018).

Internationally, some efforts have been made to develop cooking emission inventories. High-resolution emission datasets have been established for small-scale regions, such as Greater Athens in Greece and the Red River Delta in Vietnam, through field surveys and measurements (Fameli et al., 2022; Huy et al., 2021). However, at larger scales (e.g., national or global), cooking sources are often omitted from anthropogenic emission inventories or only roughly estimated using uniform emission factors and simplistic statistics like food supply or meat consumption (Huang et al., 2023; Saha et al., 2024). These methods and data are difficult to apply to China because, as mentioned above, cooking inventories in China require localized emission factors and estimation methods that explicitly consider regional differences.

Domestic inventories also exhibit the characteristic of being "precise at small scales but coarse at large scales," making it difficult to balance accuracy and breadth (Cheng et al., 2022; Jin et al., 2021; Liang et al., 2022; Wang et al., 2018a). These limitations are mainly due to the difficulty in obtaining high-quality data, particularly activity level data, over large spatiotemporal scales and at fine spatial resolutions. Some studies have collected key data for emission calculations by cuisine-specific emission factor testing, door-to-door surveys of restaurants and online fume monitoring systems, and thereby established high-resolution inventories of single years in cities or districts such as Beijing, Shanghai, and Shunde (Lin et al., 2022b; Wang et al., 2018b, 2018a; Yuan et al., 2023). These studies have provided valuable localized basic data for China's cooking emission inventories. However, obtaining accurate cooking activity data (e.g., restaurant numbers) remains challenging at larger temporal and spatial scales. Traditional China's national cooking emission inventories either use simplistic statistical data (such as population and catering consumption expenditure) as proxies for activity levels, or linearly extrapolate the activity levels of one city to other areas based on these simple statistics (Cheng et al., 2022; Jin et al., 2021; Liang et al., 2022; Wang et al., 2018a). These simplifications and linear assumptions result in high uncertainties and low spatial resolution. Recent studies have more accurately estimated national cooking emissions based on data from digital maps or catering service platforms (Li et al., 2023; Zhang et al., 2024b). However, these inventories are limited to recent years, as they rely on newly developed data platforms.

Apart from lacking accuracy and breadth, another limitation of existing cooking emission inventories is their limited pollutant coverage. Previous studies on cooking emissions primarily focused on $PM_{2.5}$ (whose organic component is primary organic aerosol, POA) and volatile organic compounds (VOCs) (Jin et al., 2021; Wang et al., 2018a, 2018b). However, recent advancements in the framework for organic compounds in the full volatility range (including VOCs, intermediate-volatility organic compounds (IVOCs), semi-volatile organic compounds (SVOCs), and organic compounds with even lower volatility (xLVOCs)) have revealed the previously overlooked significant contributions of I/SVOCs to secondary organic aerosols (SOAs) (Chang et al., 2022; Li et al., 2024b; Zhang et al., 2021). Although our latest study has supplemented the inventory with organics in the full volatility range (Li et al., 2023), the emissions for certain highly toxic pollutants of particular concern emitted from cooking- notably ultrafine particles (UFPs) and polycyclic aromatic hydrocarbons (PAHs) -

remain lacking (Chen and Zhao, 2024; Jørgensen et al., 2013; Lachowicz et al., 2023; Lin et al., 2022a). This gap limits our comprehensive assessment of the environmental and health risks associated with cooking emissions.

In recent years, machine learning has been widely applied in atmospheric pollution research due to its powerful capability to process large-scale spatiotemporal datasets and capture complex nonlinear relationships within them (Liu et al., 2023; Prodhan et al., 2022a; Zhang and Zhao, 2024; Zheng et al., 2021). Models such as Random Forest (RF), eXtreme Gradient Boosting (XGBoost), and Deep Neural Networks (DNN) have demonstrated strong performance in predicting pollutant concentration time series and identifying spatial distributions (Chen et al., 2024; Prodhan et al., 2022b; Ren et al., 2022; Wu et al., 2024; Xu et al., 2023). Ensemble machine learning models further achieve better and more stable results by combining predictions from individual base models (Liu et al., 2023; Ren et al., 2022). They can help supplement sparse datasets, serving as an effective alternative for obtaining key data that would otherwise be computationally expensive or inaccessible to collect (Ren et al., 2022; Shi et al., 2024; Xiao et al., 2018). When integrating machine learning with SHapley Additive exPlanations (SHAP) additivity algorithm, the key factors for the predictive target and their influence patterns can be identified (Hou et al., 2022; Yang et al., 2023a). Most importantly, these approaches hold significant potential to address key challenges in cooking emission inventory development. Where conventional activity data at large spatiotemporal scales are unavailable, ensemble machine learning models can predict long-term, high-resolution activity levels by capturing complex relationships between cooking activities and fundamental socioeconomic indicators. Coupled with SHAP analysis, they can provide insights into how socioeconomic factors influence emission trends. However, such efforts have not yet been made.

In summary, limited by the difficulty in obtaining high-quality activity data, there is currently a lack of an accurate, long-term, high-resolution national cooking emission, and existing inventories remain deficient in their coverage of important toxic pollutants such as PAHs and UFPs. This hinders studies on $PM_{2.5}$ modeling, source apportionment, and health risk analysis. In this study, we use machine learning models to overcome the limitations of data acquisition and driving force analysis, while also expanding the range of pollutants covered in the emission inventory. Specifically, we employ an ensemble of four preferred machine learning algorithms to estimate long-term, high-spatial-resolution cooking activity data. This ensemble model integrates the strengths of the four base models—RF, XGBoost, Multilayer Perceptron Neural Network (MLP), and DNN—enabling it to accurately predict cooking activity levels across various Chinese counties based on statistical indicators related to population, economy, and catering industry. We validate the model's generalizability and transferability using unseen testing data sets. By further combining advanced emission factors and pollution control data, we estimate the emissions of various pollutants (including organics in the full volatility range, $PM_{2.5}$, UFPs, and PAHs) from commercial, residential, and canteen cooking at the county level from 1990 to 2021. Finally, we apply the one-factor-at-a-time method to analyze the key drivers of national cooking emissions, while using the SHAP algorithm to identify the key influential factors for county-level emissions. This provides essential data and new insights for studies of the impact of cooking emissions on air pollution and human health, and helps to formulate targeted emission control policies.

We expect to achieve breakthroughs in spatiotemporal resolution, pollutant coverage, and emission estimation accuracy. This study represents the first long-term (nearly 31 years) high-resolution (county-level) inventory, whereas existing national inventories were mostly limited to single years or recent years at provincial resolution. Besides, our study covers key pollutant categories from cooking emissions, including organics in the full volatility range, PAHs, and UFPs that were not included in other national inventories. In terms of estimation accuracy, we adopt cuisine-specific emission factors, consider dynamically changing purification facility installation proportions (PFIPs) driven by provincial policies, and use precise county-level activity data to calculate emissions more accurately and better reflect regional differences. Finally, this study will provide important data and new perspectives for researching the impacts of cooking emissions on air pollution and human health, facilitating the development of targeted emission control policies.

## 2 Data and Method

The calculation method for emissions of the three sectors of cooking (commercial cooking, residential cooking, and canteen cooking) is based on Li et al., (2023b), as shown in Eq (1):

$$E = A \times [EF \times y + EF^{'}(1 - y)] \tag{1}$$

where $A$ represents the activity level, $EF$ and $EF^{'}$ are the controlled and uncontrolled EFs for a certain pollutant, and $y$ is the PFIPs.

Fig. 1 illustrates the workflow of activity level modeling, emission estimation, and driver analysis in this study. We first gather historical annual statistical data related to population, economy, and catering industry as predictive variables, and collect existing high-resolution cooking activity levels as response variables. All data is standardized to the resolution of county level, ensuring that the sample set used for modeling is rich, diverse, and of high spatial resolution. Then, we integrate four machine learning algorithms - RF, XGBoost, MLP, and DNN - which are selected for their superior predictive performance and complementary strengths, to develop predictive models for cooking activity levels across three sectors: commercial, residential, and canteen cooking. The reliability of the model is validated on unseen testing data sets. The activity levels predicted by the model, combined with emission factors and the PFIPs, can yield historical county-level cooking emissions. Finally, we apply the one-factor-at-a-time method to analyze the key drivers of national cooking emissions, whereas the SHAP algorithm is used to evaluate the relative importance of features for county-level emissions.

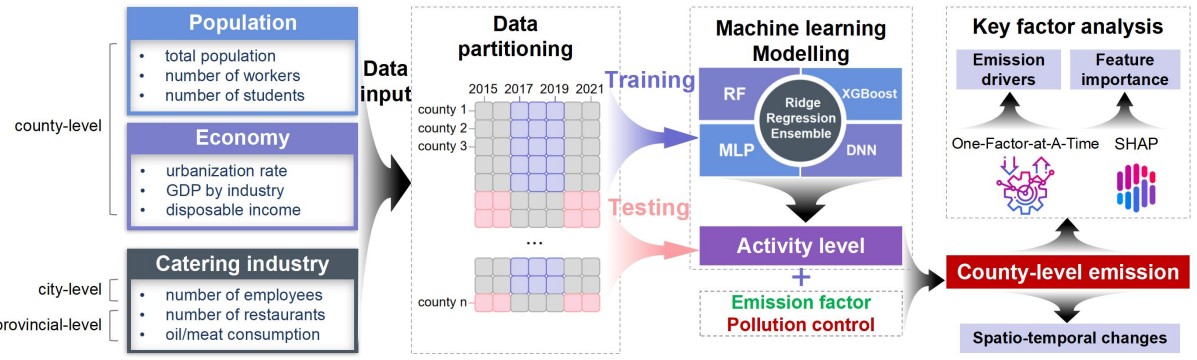

**Figure 1: Schematics of the model developed in this study including model development, emission calculation, and key factor analysis.**

## 2.1 Data acquisition and processing

To obtain long-term, high-resolution national emissions, it is important to acquire the nationwide activity level data that spans extended periods and maintains fine spatial resolution (such as county-level, or at least municipal-level). However, this is a highly challenging task, especially before the year 2000, when a significant amount of data was missing. Fortunately, we can leverage the powerful data imputation and predictive capabilities of machine learning to overcome this challenge. Specifically, the activity levels for commercial, residential, and canteen cooking are the annual total fume volume, annual

total household edible oil consumption, and the annual total number of meals served in canteens, respectively. We develop predictive models based on machine learning algorithms that only use easily accessible statistical data to estimate these county-level activity levels (as discussed in Section 2.2).

We collect 14 statistical indicators related to population, economy, and catering industry from 1990 to 2021 for modeling and predicting. The types, sources and initial resolution of all statistical data can be accessed in Table S1. Population-related

variables include population, the number of employees in enterprises, and the number of students in primary school and middle school. Economy-related variables encompass urbanization rate, total gross domestic product (GDP), GDP of primary, secondary, and tertiary industries, and per capita disposable income. Variables related to the catering industry include household per capita oil consumption, household per capita meat consumption, the number of chain restaurants, and the number of employees in the catering and accommodation industry. These data, mostly at the county-level resolution,

primarily originate from statistical yearbooks (National Bureau of Statistics of China, 2022a, c, b). These long-term datasets are preprocessed to meet the requirements of machine learning by imputing missing values using inverse distance weighting, K-nearest neighbor methods, and allocation of higher-order statistical data (Murti et al., 2019; Sree Dhevi, 2014). Given the changes in China's county administrative divisions over the past 31 years (Yu et al., 2018), we trace the renaming, merging, and splitting events of counties according to official government reports, mapping the data of each year to the county

administrative system of 2020 (a total of 2848 counties) to ensure continuity across years. Detailed descriptions of the spatial

mapping and data processing procedures are provided in Text S1. Additionally, we standardize the initial resolution of some variables, which may be at the provincial, municipal, or grid level (1km*1km), to the county level by allocating based on population or GDP, taking provincial averages, or using cumulative summation. We also normalized all predictor variables to a range of 0 to 1 to ensure a consistent scale.

Next, we conduct feature selection on 14 predictor variables to reduce dimensionality and minimize multicollinearity (Zhu et al., 2023). Variables were first ranked by RF model importance scores, with those scoring <5% considered for potential removal because of low importance (Alduailij et al., 2022; Ye et al., 2022). Each candidate variable was then temporarily excluded, and only those whose removal resulted in an R² decline of <1% were permanently discarded – ensuring no critical features were lost (Altmann et al., 2010; Zhu et al., 2023). Besides, we perform multicollinearity checks using the variance

inflation factor (VIF), gradually removing features with higher VIF values until all remaining features were mutually independent (all VIF values of independent variables were below 10) (Daoud, 2017; Hu et al., 2017). By removing irrelevant or redundant features in this way, we can reduce the influence of noise, decrease the risk of overfitting, enhance the model's predictive performance and generalizability, and provide clearer and more meaningful model explanations (Zhu et al., 2023).

For machine learning modeling, the dataset needs to be partitioned into the training data set and the testing data set. During

the data partitioning, we implement strict data leakage management to ensure that information from the testing data set would not be used during training, thus guaranteeing an accurate model evaluation (Nayak and Ojha, 2020; Zhu et al., 2023). The response variables available for modeling and testing, namely high-resolution cooking activity levels, are limited to the years 2015 to 2021 (Li et al., 2023b). This gives us data samples for seven years, with 2848 counties each year. Given the significant similarity in data for the same county across different years, we bundle data samples from different years for the

same county during the data partitioning. As shown in the second column of Fig. 1, we use data from 70% of the counties from 2017 to 2019 (totaling 5982 samples) for training to establish the underlying relationships between input factors and the prediction target. Additionally, we use data from the remaining 30% of the counties in the years 2015, 2016, 2020, and 2021 (totaling 3416 samples) as the testing data set to validate the model. Under this partitioning strategy, data from the same county only appears in either the training data set or the testing data set, ensuring that the model can effectively generalize

and be tested on unseen datasets, thereby demonstrating the model's transferabilty across different times and locations (Nayak and Ojha, 2020; Zhu et al., 2023). Notably, the modeling data only covers the period from 2015 to 2021, as earlier data are unavailable. This may introduce some bias when backcasting activity levels for earlier periods. We further validate the backcasted activity levels and evaluate their uncertainties. Modeling and validation of the machine learning model are described in detail in section 2.2.

After obtaining activity levels through the machine learning model, we further collect data for Eq (1) to calculate cooking emissions. As for the EF, we consider various types of pollutants of concern emitted from cooking activities, including organics in the full volatility range (VOCs, SVOCs, IVOCs, and xLVOCs), $PM_{2.5}$, UFP, and PAHs (encompassing gaseous PAHs, particulate PAHs, and benzo[a]pyrene toxic equivalent quantity ($BaP_{eq}$)). The organic EFs in the full volatility range

are sourced from Li et al., (2023b). The EFs for PM$_{2.5}$ are calculated as POA/81.5% (Li et al., 2023b), where POA represents the particulate fraction of organics in the full volatility range. The EFs for UFPs are derived from the literature (Chen et al., 2017, 2018; Géhin et al., 2008; Kim et al., 2024; Zhang et al., 2010). The EF of gaseous and particulate PAHs are mainly sourced from simultaneous gas-particle testing in multiple studies (Chen et al., 2007; Feng et al., 2021; Li et al., 2003, 2018; Lin et al., 2022a; Saito et al., 2014; Ye et al., 2013). We consider 16 priority PAHs and 5 non-priority PAHs commonly found in cooking emissions. Their BaP$_{eq}$ are calculated based on the recommended toxic equivalency factors (TEFs) suggested in the literature (Greim, 2008; Larsen et al., 1998; Malcolm et al., 1994; Nisbet et al., 1992). The molecular information and recommended TEF values for all PAH species considered in this study are listed in Table S2. The specific values and sources of EFs for various pollutants are listed in Table S3.

As for PFIPs, we have established a grading standard for provincial catering emission control stringency and corresponding PFIPs based on field surveys (Li et al., 2023b). By collecting provincial-level catering pollution control policies and considering their implementation timelines and transition periods, we can obtain dynamically changing PFIPs driven by provincial-level control policies. In this study, we also applied this method to estimate PFIPs for 1990-2021 (See Table S4-5 for the PFIPs results over the years).

## 2.2 Establishment and optimization of ensemble machine learning model

Ensemble methods of machine learning have recently been increasingly applied in the large-scale spatiotemporal estimation of atmospheric pollution (Yang et al., 2023b; Zhu et al., 2022). These methods enhance prediction accuracy and robustness by combining the forecast results from multiple base models and reducing the risk of overfitting. In this study, we establish an ensemble prediction model for cooking activity levels by integrating four machine learning algorithms - RF, XGBoost, MLP, and DNN. These four models are selected because they exhibit superior performance in predicting activity levels (as discussed in Section 3.1), and each of them possesses unique strengths, as discussed below.

RF and XGBoost are both ensemble learning algorithms based on decision trees. RF improves accuracy and generalization by combining multiple independent decision trees, making it suitable for handling high-dimensional data (Liu et al., 2023; Segal, 2004). Its advantage lies in the effective reduction of overfitting through random feature selection (Wu et al., 2024). XGBoost, as an efficient gradient-boosting decision tree method, also reduces overfitting by introducing regularization and has a high execution speed, making it suitable for processing large-scale datasets (Chen and Guestrin, 2016). While these tree-based algorithms provide stable predictions and good interpretability, they may have limited extrapolation capabilities (Wang et al., 2023). To address this, we introduce MLP and DNN, two deep learning algorithms, to enhance the model's applicability. MLP, a fundamental deep learning model with a multi-layer structure, can capture complex nonlinear trends in data and can infer patterns beyond the training data range, with lower computational requirements compared to other deep learning models (Pinkus, 1999). DNN, on the other hand, captures advanced abstract features in complex data through

deeper network structures, offering powerful feature learning and generalization capabilities (Zhang et al., 2016). However, both MLP and DNN may face the challenge of overfitting (Pinkus, 1999; Zhang et al., 2016), which can be mitigated by integrating them with RF and XGBoost.

To combine the advantages of these four models, we use ridge regression as the integrator to build an ensemble machine learnling model (McDonald, 2009). Ridge regression is chosen for its ability to balance model complexity and generalization through regularization, which helps prevent overfitting (Ebrahimi et al., 2024; McDonald, 2009). Furthermore, as validated in Text S2 and Table S6, ridge regression demonstrates a favorable balance between performance and computational efficiency when compared to other fusion strategies. Specifically, the predictions from the base models serve as new features input into the ridge regression model, which then determines how to effectively combine these predictions (Carneiro et al., 2022). This approach allows us to leverage the strengths of each model: the interpretability and stability of RF and XGBoost, and the ability of MLP and DNN to capture complex nonlinear patterns. By integrating these models, we aim to achieve a more robust and accurate prediction model that can handle diverse data scenarios (Carneiro et al., 2022).

Due to variations in influencing factors and mechanisms within different cooking emission sectors, we develop an ensemble model for commercial, residential, and canteen cooking, respectively. For each sector's training data set, models are trained using 10-fold cross-validation to ensure that their predictive capabilities are not influenced by specific data subsets (Santos et al., 2018). Moreover, a grid search is conducted on the hyperparameters of each base machine learning model and the ridge regression model to identify the optimal hyperparameter combination that maximizes overall predictive performance (Belete and Huchaiah, 2022; Lou et al., 2024)

**2.3 Model validation and comparison**

After completing the modeling, we apply the models to the unseen testing data sets and evaluate their predictive performance using various statistical metrics. The validation metrics include the coefficient of determination ($R^2$), root mean square error (RMSE), and mean absolute error (MAE). Their calculation formulas are as follows:

$$R^2 = 1 - \frac{\sum_{i=1}^{n}(Obs_i - Pred_i)^2}{\sum_{i=1}^{n}(Obs_i - \text{MeanObs})^2} \tag{2}$$

$$RMSE = \sqrt{\frac{\sum_{i=1}^{n}(Obs_i - Pred_i)^2}{n}} \tag{3}$$

$$MAE = \frac{1}{n}\sum_{i=1}^{n}|Pred_i - Obs_i| \tag{4}$$

where $Obs_i$ represents the actual values (i.e., the activity levels obtained from accurate calculations); $Pred_i$ refers to the model-predicted activity levels; MeanObs is the average of all $Obs_i$; n is the number of samples in the testing data sets.

To demonstrate the superiority of our ensemble model, we also compare its predictive performance with the above-mentioned four individual machine learning models and five advanced traditional statistical models, including multiple linear regression, non-negative least squares regression, generalized linear models with exponential link, Poisson regression, and power function regression (Frome, 1983; Jansson, 1985; Myers and Montgomery, 1997; Slawski and Hein, 2013; Uyanık and Güler, 2013).

## 2.4 National driver analysis and county-level feature importance analysis of cooking emissions

Based on the model-predicted cooking activity levels, the EFs applicable at all times, and PFIPs that can be extrapolated to any time, we theoretically can estimate the cooking emissions in various scenarios (such as different population conditions, economic circumstances, and pollution control intensities). In this study, we first obtain the emissions of three cooking sectors in each county from 1990 to 2021. Further, we can conduct sensitivity analysis on emissions by adjusting various influencing factors (input features of the ensemble model and PFIPs). Since EFs are static data that do not change across different years, we do not consider their impact. We first pay attention to the national total emissions, using the one-factor-at-a-time method (Zhang et al., 2018) to illustrate the sensitivity of each factor to emission variations. We divide the years from 1990 to 2021 into several periods. For a given period, we sequentially adjust the value of a single factor from the initial value at the beginning of the period to the final value at the end of the period. The difference between the emissions before and after the adjustment is considered as the contribution of that factor to the change in emissions during that period. This enables us to quantify the contributions of each factor to emissions in different periods.

Notably, due to interaction effects among variables, different adjustment sequences may yield distinct driver decomposition results. To enhance the robustness, we mitigate sequence-dependent bias by incorporating the averaged results from multiple adjustment sequences. Specifically, we first identify the top five variables with the largest individual impacts on emission changes in each period. We then exhaustively examine all possible permutations (120 in total) of these five variables and apply the adjustments accordingly. For variables ranked sixth and below, adjustments are made in descending order of their individual impacts, without considering further permutations, as these variables contribute minimally to emission changes (<2% annually), and exhaustive permutations would be computationally prohibitive. Finally, we calculate the average contribution of each factor across all considered sequences. While this improved approach cannot completely decouple the independent contributions of all variables, it significantly reduces order-dependent biases and enhances the reliability of the driver decomposition.

The dominant factors associated with emission changes for counties at different development stages are also worth elucidating, which are crucial for understanding the current and future trends in cooking emissions, and for the targeted development of control strategies. We employ the SHAP algorithm (Lundberg and Lee, 2017) to quantify the impact of each factor on cooking emissions in different counties. These factors include those features related to population, economy, and catering industry that are input to the activity level prediction model, as well as the EF and the PFIP. The SHAP algorithm is

based on cooperative game theory (Jiménez-Luna et al., 2020; Lundberg and Lee, 2017). By including or excluding a variable from all possible subsets of the remaining variables, the model is retrained to calculate the difference in predicted values in two scenarios, referred to as SHAP values. The magnitude of SHAP values quantifies the specific contribution of each feature to the model's predictions. A positive value indicates that the feature raises the predicted result relative to the baseline, while a negative value signifies a reduction in the predicted result(Hou et al., 2022; Zhu et al., 2023).

## 3 Results

### 3.1 Performance comparison of the models

We first train five traditional statistical models, four individual machine learning models, and our ensemble model using the training data set. The performance of each model on the training data set is shown in Table S7. Then, all models are applied to an unseen testing data set (detailed dataset partitioning is described in Section 2.1) to assess their performance in predicting the activity levels of three cooking sectors. The predictive performance of all models for activity levels of three cooking sectors on the testing data set is shown in Table 1 and Fig. S1-3. To enhance clarity, we scale the units for the three activity levels: the activity levels for commercial, residential, and canteen cooking are represented respectively as annual total fume volume (unit: $10^9$ m$^3$ fume), annual total household edible oil consumption (unit: kt oil), and annual total number of meals served in canteens ($10^6$ meals). We also present the predictive performance of the best statistical models, the best individual machine learning models, and the ensemble machine learning model for the three cooking sectors in Fig. 2(a).

**Table 1: The values of validation metrics of all models for activity levels of three cooking sectors on the testing data set.**

| Model | Commercial cooking | | | Residential cooking | | | Canteen cooking | | |
|---|---|---|---|---|---|---|---|---|---|
| | $R^2$ | RMSE ($10^9$ m$^3$) | MAE ($10^9$ m$^3$) | $R^2$ | RMSE (kt) | MAE (kt) | $R^2$ | RMSE ($10^6$ meals) | MAE ($10^6$ meals) |
| Multiple linear regression | 0.718 | 25.618 | 16.199 | 0.936 | 1.078 | 0.625 | 0.955 | 5.697 | 3.202 |
| Non-negative least squares regression | 0.617 | 29.857 | 18.910 | 0.898 | 1.368 | 0.953 | 0.955 | 5.750 | 3.172 |
| Generalized linear models with exponential link | 0.625 | 29.548 | 17.777 | 0.348 | 3.455 | 2.649 | 0.496 | 19.157 | 14.666 |
| Poisson regression | 0.454 | 35.673 | 19.737 | 0.056 | 4.156 | 2.947 | 0.278 | 22.940 | 16.294 |
| Power function Regression | 0.772 | 23.055 | 10.599 | 0.965 | 0.804 | 0.406 | 0.950 | 6.044 | 3.085 |
| RF | 0.835 | 19.589 | 10.173 | 0.979 | 0.618 | 0.155 | 0.958 | 5.545 | 3.109 |
| XGBoost | 0.807 | 21.224 | 11.582 | 0.971 | 0.726 | 0.277 | 0.958 | 5.561 | 3.037 |
| MLP | 0.856 | 18.316 | 9.867 | 0.972 | 0.714 | 0.185 | 0.970 | 4.675 | 1.750 |
| DNN | 0.866 | 17.644 | 8.355 | 0.970 | 0.738 | 0.231 | 0.969 | 4.764 | 2.360 |
| Ensemble machine learning model | 0.892 | 15.834 | 7.968 | 0.989 | 0.455 | 0.109 | 0.973 | 4.447 | 1.832 |

According to Table 1, validation metrics indicate that machine learning models greatly outperform the best traditional statistical models, with the ensemble machine learning model even surpassing the best individual machine learning models. Among statistical models, multiple linear regression has moderate performance, but it is prone to predicting negative values, which do not correspond to real-world cooking activity. Besides, among the other four non-negative predictive models, power function regression performs best for predicting commercial cooking and residential cooking, while non-negative least squares regression performs best for predicting canteen cooking. Generalized linear models with exponential links and Poisson regression perform poorly in most cases.

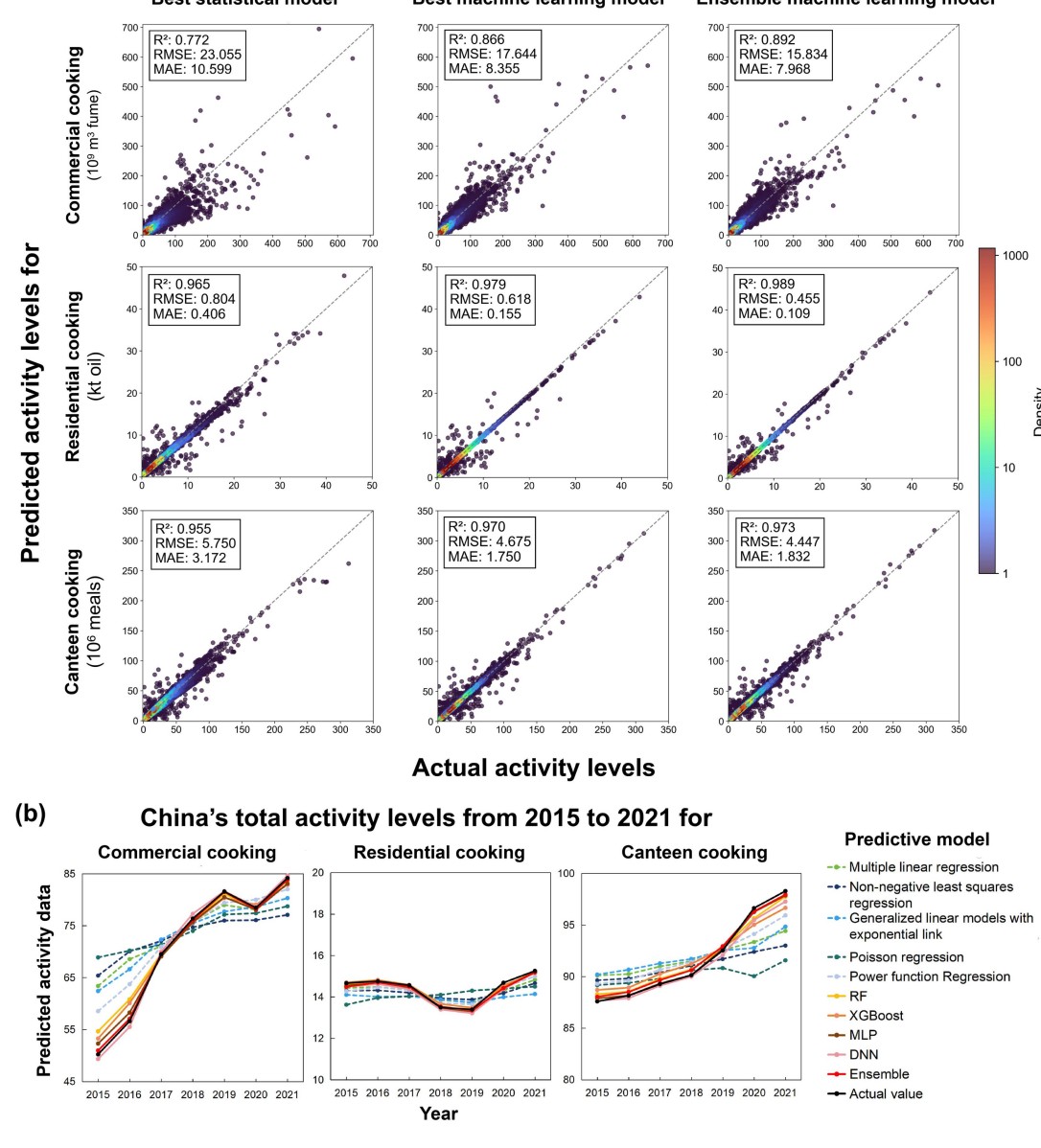

**Figure 2: Comparison of statistical models, individual machine learning models, and ensemble machine learning models:** (a) Scatter plots comparing the actual and predicted values of the best statistical model, best individual machine learning model, and ensemble machine learning model for the activity levels in three cooking source sectors in China, with each point representing the activity level in a county from the testing data set. (b) Predicted and actual values of Chinese total activity levels of the three cooking sectors for each year from 2015 to 2021. The black line represents the actual values. Lines of other colors represent model predictions, where the solid lines are for machine learning model predictions, and dashed lines are for predictions from traditional statistical models.

Machine learning models tend to have better predictive capabilities than the traditional statistical models. Among the five machine learning models, we find that ensemble machine learning models consistently perform the best, with $R^2$ values of 0.892, 0.989, and 0.973 for commercial cooking, residential cooking, and canteen cooking activity levels, respectively. RMSE and MAE metrics of the ensemble models are also relatively low. The superiority of validation metrics implies that the ensemble model can effectively depict the relationship between indicators related to statistic indicators and cooking activity levels. Moreover, the overall performance of individual machine learning models is also satisfactory. Specifically, for commercial cooking and canteen cooking, which are influenced by complex factors, the performance of the two deep learning models is superior, as they are more adept at capturing complex nonlinear relationships. On the other hand, for residential cooking, whose influencing factors are relatively simple and clear, the performance of RF is better than that of deep learning models, possibly because it can effectively prevent overfitting. Finally, the ensemble models can exploit complementary advantages, reduce the uncertainties of single models, and achieve performance maximization.

We also review the predictive performance of all models on the Chinese total activity levels of the three cooking sectors for each year from 2015 to 2021, as shown in Fig. 2(b). Although the training data set was randomly sampled from counties only from 2017 to 2019, the machine learning models (represented by the solid line) demonstrate a robust ability for generalization and extrapolation. They accurately capture the Chinese total activity level trends of the modeling years (2017-2019) and extend to historical years (2015-2016) and future years (2020-2021), whereas traditional statistical models (represented by the dashed lines) often fail to accurately reproduce the changes in total activity levels.

While the model demonstrates good performance for near-term extrapolation, greater uncertainty may exist when backcasting to earlier periods, which include more underdeveloped counties. To evaluate this, we conduct sensitivity analyses by training on the top 70% GDP-ranked counties and testing on the bottom 30%. For commercial catering (the most complex case, as shown in Fig. S4), the ensemble models test-set $R^2$ remained robust ($R^2$=0.719), outperforming the best statistical models ($R^2$=0.523). For real-world backcasting of historical data, the $2015-2021$ training data already include some less-developed regions that can represent early-stage conditions, mitigating extreme extrapolation risks. Additionally, we further validate the historical trends of predicted activity data based on the limited available historical data (Fig. S5).

From 1990 to 2021, the growth rate of commercial cooking activity levels was intermediate between population growth and tertiary GDP. The temporal evolution resembles that of the chain restaurant number (slow early growth followed by acceleration), though the chain restaurant number is more stable as they exclude small independent restaurants. We also incorporate chain restaurant revenue data (available since 2004), which corroborates the fluctuations in our predictions including the post-2015 rapid growth driven by food delivery platforms and the 2020 pandemic-driven decline (Maimaiti et al., 2018; Zhao et al., 2021). Therefore, while temporal extrapolation may introduce biases (uncertainties are quantified in Section 3.3), our multi-pronged validation demonstrates the reasonableness and optimality of our backcasted estimates.

## 3.2 Long-term county-level cooking emissions

After verifying the reliability and superiority of the ensemble model, we utilize it to predict precise county-level activity data at a broad spatial and temporal scale, and further obtain county-level cooking emissions in China from 1990 to 2021. Each county's annual emissions inventory for organics (hereafter representing the organic compounds in the full volatile range), $PM_{2.5}$, UFPs, and PAHs is available at the repository (https://doi.org/10.6084/m9.figshare.26085487) (Li et al, 2025). Considering that organics have significant impacts on atmospheric pollution, particularly OA pollution, and that various pollutants share the same activity levels leading to similar spatial and temporal distributions, we primarily focus on organic compounds in the following discussion.

Fig. 3 provides high-resolution spatial distribution maps of cooking organic emissions in China from 1990 to 2021, while Fig. S6 provides the sector-specific spatial distributions of cooking organic emissions for a representative year (2021). We also provide a map of the Chinese provinces (Fig. S7) for reference to the location of the emissions mentioned below. In 1990, cooking organic emissions were mainly distributed in densely populated areas such as the North China Plain (including Beijing, Tianjin, Hebei, Henan, and Shandong), the Middle-Lower Yangtze Plain (including Hubei, Hunan, Anhui, Jiangxi, Jiangsu, and Zhejiang), and Sichuan Basin (including Sichuan and Chongqing). Besides, emission hotpots were often observed in the core urban areas of provincial capitals. Over time, the national total organic emissions have generally increased, and high-emission areas have expanded. By 2021, many counties in eastern China, especially along the southeast coast, exhibited extensive high emissions. The North China Plain region, the Yangtze River Delta, the Pearl River Delta, and the Sichuan-Chongqin region became the four key emission zones, contributing 20.2%, 19.9%, 8.63%, and 7.98%, respectively, of the nation's total cooking organic emissions.

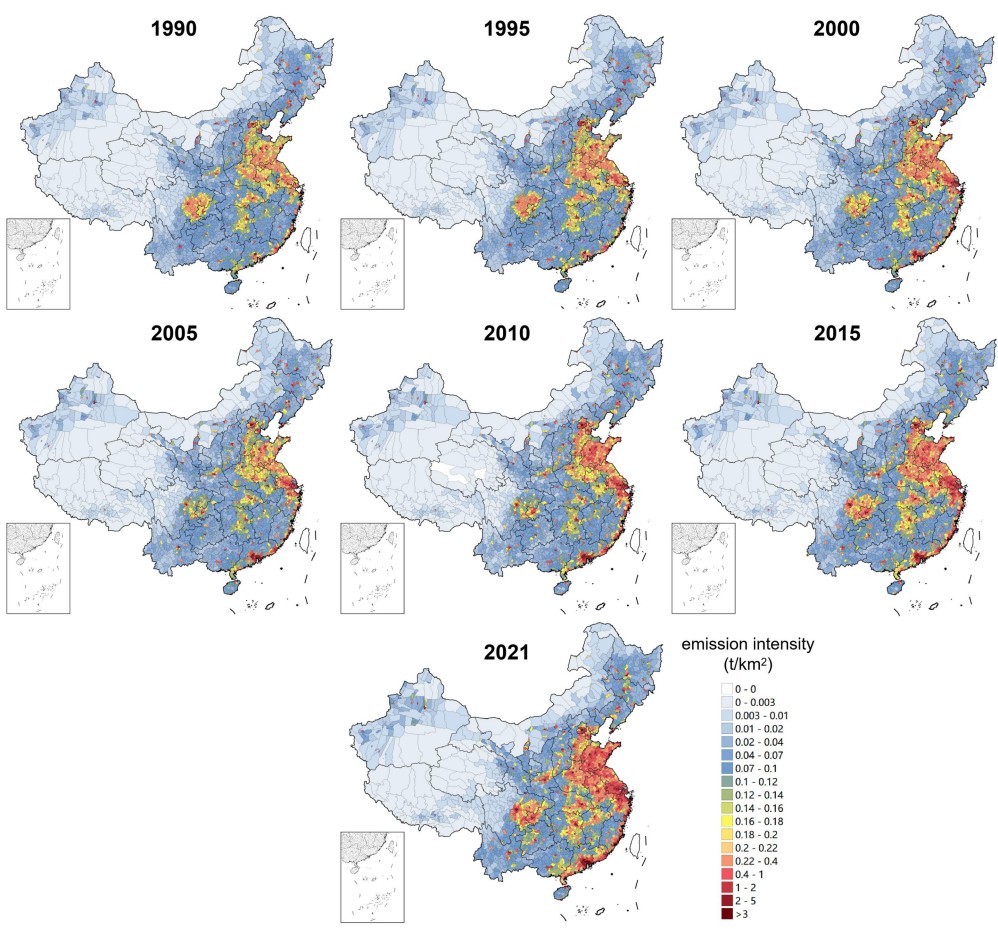

**Figure 3: The spatial distribution of nationwide county-level cooking organic emission intensity from 1990 to 2021.**

In summary, cooking emissions are concentrated in densely populated and economically developed areas. For example, in 2021, there was a strong correlation between county population size and cooking emissions, with an $R^2$ of 0.873 for the emissions and population of 2848 counties in 2021. Notably, the top 30% of counties by population (as shown in Fig. S8(a)) accounted for 62.3% of the total national cooking organic emissions. These counties cover only 14.5% of China's total land

area but support 59.9% of the country's population. This finding indicates that, when formulating control strategies, these densely populated counties should be prioritized to enhance pollution control efficiency and effectively reduce the health risks associated with cooking emissions. From 1990 to 2021, the proportion of total national emissions contributed by the top 30% of counties by population increased from 49.6% to 62.3%, suggesting that cooking emissions in densely populated counties have grown faster than in other areas, necessitating stricter pollution control measures. Additionally, cooking

emissions are also correlated with local GDP, although this correlation is weaker than with population, with an $R^2$ of 0.563

for emissions and GDP across all counties in 2021. The top 30% of counties by GDP (as shown in Fig. S8(b)) accounted for 55.9% of the total national cooking organic emissions. Besides, we further analyze the spatiotemporal trends of emission distribution and their underlying socioeconomic drivers in Section 4.1.

Fortunately, in these densely populated and economically developed areas (Fig. S8), where emissions are typically high, our county-level emissions inventory achieves a very high spatial resolution. Compared to traditional provincial inventories, our fine-grained inventory is more capable of accurately studying the impact of cooking emissions on air pollution and human health. Specifically, our inventory may update the understanding of $PM_{2.5}$ sources. Combining the full-volatility organic emissions inventory (excluding the cooking source) developed by Zheng et al. (2023), we find that cooking emissions are significant sources of I/SVOC emissions in densely populated counties. In 2019, for counties within the top 30% of population density (as shown in Fig. S8(c)), cooking emissions can account for an average of 20.1% of IVOCs and 38.5% of SVOCs emitted from all anthropogenic sources, and the maximum contribution of cooking emissions to total IVOC and SVOC emissions in these counties even reached 52.9% and 88.4%, respectively. Given the high formation potential for SOA of I/SVOCs emitted from cooking (Yu et al., 2022), the contribution of cooking organic emissions to $PM_{2.5}$ and their hazards on human health could be substantial. However, if considering only national or provincial emissions, the contribution of cooking emissions to the total IVOC emissions and total SVOC emissions are both less than 16%, potentially leading to an underestimation of the importance of the cooking source.

**3.3 Trends of national total cooking emissions**

Fig. 4 illustrates the long-term trend of national cooking emissions of organic compounds in the full volatility range from 1990 to 2021. The total cooking organic emissions in China exhibit an overall increasing trend, rising from 517 (272-828, 95% confidence level) kt/yr in 1990 to 997 (530-1590) kt in 2021. The uncertainty ranges are determined through Monte Carlo simulations referencing previous studies (Chang et al., 2022; Nan Li, 2017), incorporating cumulative biases introduced by extrapolating historical emissions using limited training data (see Text S3 for details). Notably, there were slight decreases in total organic emissions after 2001 and after 2013, attributed to the implementation of crucial control policies. In 2001, the issuance of the *Emission Standards of Catering Oil Fume* (GB 18483-2001) (State Environmental Protection Administration of China, 2001) marked the first significant attention of the Chinese government to cooking emission control. It imposes requirements on the concentration of oily fumes emitted by restaurants and the removal efficiency of the purification facilities, which has contributed to the reduction of emissions (State Environmental Protection Administration of China, 2001). Furthermore, the release of *the Action Plan for the Prevention and Control of Air Pollutants* in 2013 pushed provinces to comprehensively strengthen air pollution control (CPGPRC, 2013), leading to a corresponding enhancement of the catering industry's regulation in many regions. Additionally, the downturn observed in the 2020 emission was brought about by the lockdown measures implemented due to the COVID-19 pandemic.

As for source apportionment, the cooking organic emissions mainly come from commercial cooking and residential cooking. Commercial cooking emissions show an overall upward trend, with some slight fluctuations due to its high sensitivity to external factors such as pollution control policies and epidemic lockdowns. Commercial cooking emissions have increased from 241 kt in 1990 to 622 kt in 2021, and its share has correspondingly increased from 46.7% to 62.3%. Residential cooking emissions show an overall slow upward trend, with its share ranging between 28.3% and 37.2%. In contrast, canteen cooking emissions show an overall stable or slightly declining trend. This is possible because they mainly come from staff and student canteens, where the number of staff and students and their meal frequencies are relatively stable. However, with pollution control measures becoming stricter, this has led to a reduction in total canteen cooking emissions.

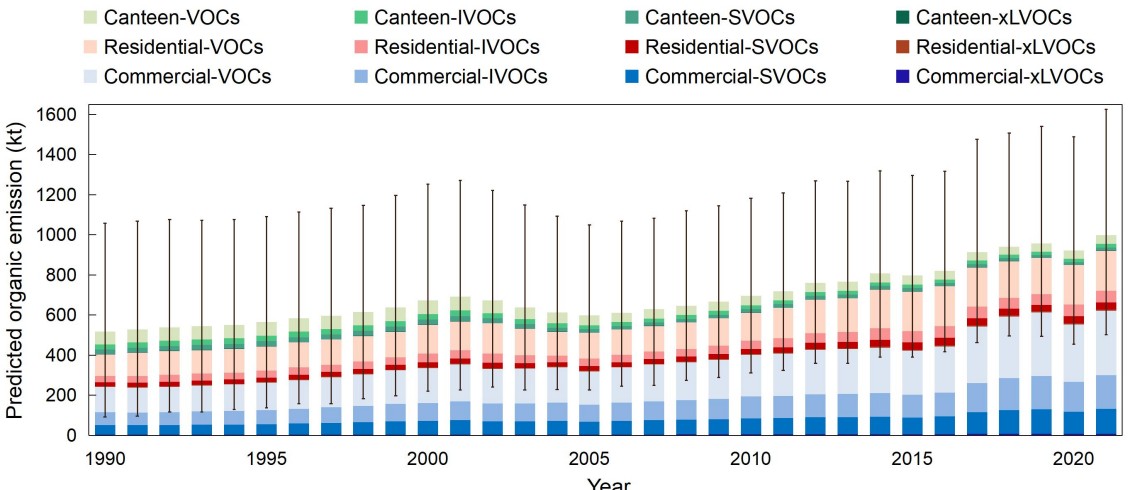

**Figure 4: Organic emissions in the four volatility ranges from the three cooking sectors from 1990 to 2021 in China.** The blue, red, and green bars represent the organic emissions from commercial cooking, residential cooking, and canteen cooking. Within each color group, the four different shades represent organic compounds of different volatility ranges. The error bars represent the uncertainty range at the 95% confidence level.

Furthermore, we also present emissions of $PM_{2.5}$, UFPs, and PAHs (including gaseous PAHs, particulate PAHs, and $BaP_{eq}$) from the three cooking sectors in China from 1990 to 2021, as shown in Fig. S9. The trends and source apportionment of $PM_{2.5}$ emissions are similar to those of organic emissions. The total $PM_{2.5}$ emissions increased from 215 kt in 1990 to 408 kt in 2021, representing a growth of 90.7%. Commercial cooking is the most significant emission source, accounting for 39.3%-57.7%, followed by residential cooking (34.8%-44.3%). The total UFP emissions increased from $3.93 \times 10^{25}$ particles in 1990 to $6.50 \times 10^{25}$ particles in 2021, with an increase of 66.0%. Commercial emissions have consistently been the largest source, maintaining a share of over 71%.

The total PAH emissions increased from 6.76 kt in 1990 to 15.8 kt in 2021, representing a growth of 134%. The $BaP_{eq}$ emissions rose from 0.359 kt in 1990 to 0.853 kt in 2021, with an increase of 137%. Additionally, the emissions of the 16 priority PAHs increased from 6.20 kt in 1990 to 14.5 kt in 2021. After supplementing the emissions inventory of the 16 priority PAHs in China (excluding cooking sources) by Wang et al. (2021), we find that cooking emissions accounted for 11.0% of the total anthropogenic emissions of priority PAHs in China in 2017, and the share may be even larger in urban areas. Among these priority PAHs, naphthalene has the highest emissions share (46.8%), followed by acenaphthylene (11.7%) and phenanthrene (10.5%). As for toxicity, dibenz(a,h)anthracene has the highest $BaP_{eq}$ emissions share (42.8%), followed by benzo(a)pyrene (36.1%). Notably, high molecular weight PAHs (containing five- to seven-ringed PAHs) accounted for only 8.2% of the emissions but contributed 85.3% of the $BaP_{eq}$ emissions due to their high toxicity. Besides, over the 31 years, gaseous and particulate PAHs accounted for an average of 78.6% and 21.4% of the total PAH emissions, respectively. Commercial cooking remained the primary emission source, contributing 74.6%-83.2% of the national PAH emissions.

### 3.4 Comparison with other studies

We compare our cooking emission inventory with other China's national cooking emission inventories (Cheng et al., 2022; Jin et al., 2021; Liang et al., 2022; Wang et al., 2018a; Zhang et al., 2024b). Most previous inventories only included pollutants such as VOC and $PM_{2.5}$ (or organic carbon (OC), a component of $PM_{2.5}$), and provided emissions for only a single year. We first compare the national total emissions for the corresponding years and pollutants with theirs, and the results are presented in Table S8. Many previous inventories underestimated emissions due to the omission of emission sources (e.g., residential cooking) or the use of simple proxy data (e.g., population, meat consumption) (Cheng et al., 2022; Jin et al., 2021; Liang et al., 2022; Wang et al., 2018a), so their total emissions are much lower than ours. The latest studies (Zhang et al., 2024b), which used data from a service platform of Chinese catering enterprises, yielded national total VOC emissions relatively close to those of our inventory, supporting the accuracy of our emission calculations. In contrast, our inventory covers a longer time range (1990–2021), comprehensive cooking sources (including commercial, household, and canteen cooking), and a wider range of pollutants (not limited to VOC and $PM_{2.5}$, but also including PAHs, UFP, etc.), which is difficult to achieve in previous studies.

Furthermore, our inventory demonstrates superior accuracy in spatial distribution. Unlike previous studies, this study precisely calculates emissions at the county level, rather than first estimating provincial-level inventories and then downscaling them to the county level (or further to the grid level) using proxy data such as population. We compare our inventory with the aforementioned latest inventory based on data from the catering service platform (Zhang et al., 2024b), which calculated the provincial emission inventory and then allocated it to the county levels based on population. We select the county-level emissions in Guangdong in 2020 as a case study for comparison, as Guangdong is a province with high cooking emissions, a large population, and a developed economy. In terms of total emissions, the Guangdong provincial

emissions from this study and Zhang's inventory are 63.2 kt and 58.6 kt, respectively (Zhang et al., 2024b), showing close agreement. Fig. 5 illustrates the emission intensity across all counties in Guangdong from the two inventories. The key difference between the two is that the emissions in our study are more concentrated in economically developed regions such as the Pearl River Delta, while the emission intensity in non-coastal areas is lower. This discrepancy arises because allocating provincial inventories to the county level based on population distribution may not fully reflect real-world conditions. In fact, some residential areas may have high population density, but dining activities are often more concentrated in commercial districts (Lin et al., 2022b). As discussed in Section 3.2, although the correlation between population and emissions is high at the county level ($R^2 = 0.873$), it is not a perfect match. In contrast, our methodology employs an effective machine learning model trained on advanced point-source cooking emission inventories (Li et al., 2023), with predictive variables related to population, economy, and catering industry. This method effectively captures the spatial distribution of comprehensive cooking activities, including information on the catering industry, residential cooking, and other factors considered in the previous advanced inventory (Li et al., 2023), thereby enabling an accurate representation of the spatial distribution of county-level cooking emissions.

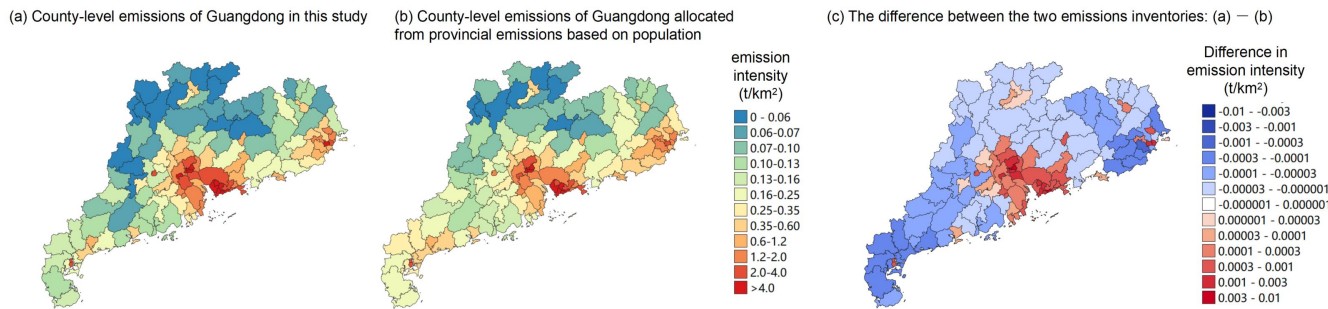

**Figure 5: A comparison of (a) county-level emissions in this study, (b) county-level emissions allocated from provincial emissions based on population in Guangdong in 2020, and (c) the difference between the two emissions inventories.**

## 4 Discussion

### 4.1 Spatiotemporal trends of county-level cooking emissions

To explore the spatiotemporal variation trends of cooking emissions in China, we obtain the changes in county-level organic emissions every 5 or 6 years through differencing, as illustrated in Fig. 6, where the red color indicates an increase in emissions during a particular period and blue represents a decrease. Our modeling explicitly incorporates key socioeconomic and policy variables such as population, urbanization rate, and PFIP, and captures their relationships with county-level cooking emissions. Therefore, the spatiotemporal variation trends of emissions can be well explained by these underlying socioeconomic and policy drivers.

Before 2000, the observed changes in emissions could primarily be explained by population and economic factors, since China had few policies for cooking emission control (Zhao, 2004). From 1990 to 1995, emissions across most counties generally increased because of economic development, but emissions in a few counties decreased probably due to population migration. The most significant emission growth was concentrated in Beijing, Shanghai, and the Pearl River Delta region - areas undergoing greatly rapid economic development and urban expansion at the time (Démurger et al., 2002; Gaubatz,

2004). Conversely, emission reductions were typically observed in adjacent areas, suggesting population redistribution toward these emerging economic centers (Fan, 2005). From 1990 to 1995, the migration-driven spatial redistribution of emissions became more pronounced between 1995-2000 (Fan, 2005). For example, Guangdong's emissions became concentrated in the Pearl River Delta, while Zhejiang and Fujian's emissions clustered in the Yangtze River Delta and coastal regions. Besides, emissions from eastern Sichuan are shifting towards Chengdu (the provincial capital of Sichuan) and

Chongqing. The migration was probably because these areas became focal points of economic reform during this period, attracting large population influx (Fang et al., 2009), such as Chongqing being designated a directly-controlled municipality in 1997 (Hong, 2004).

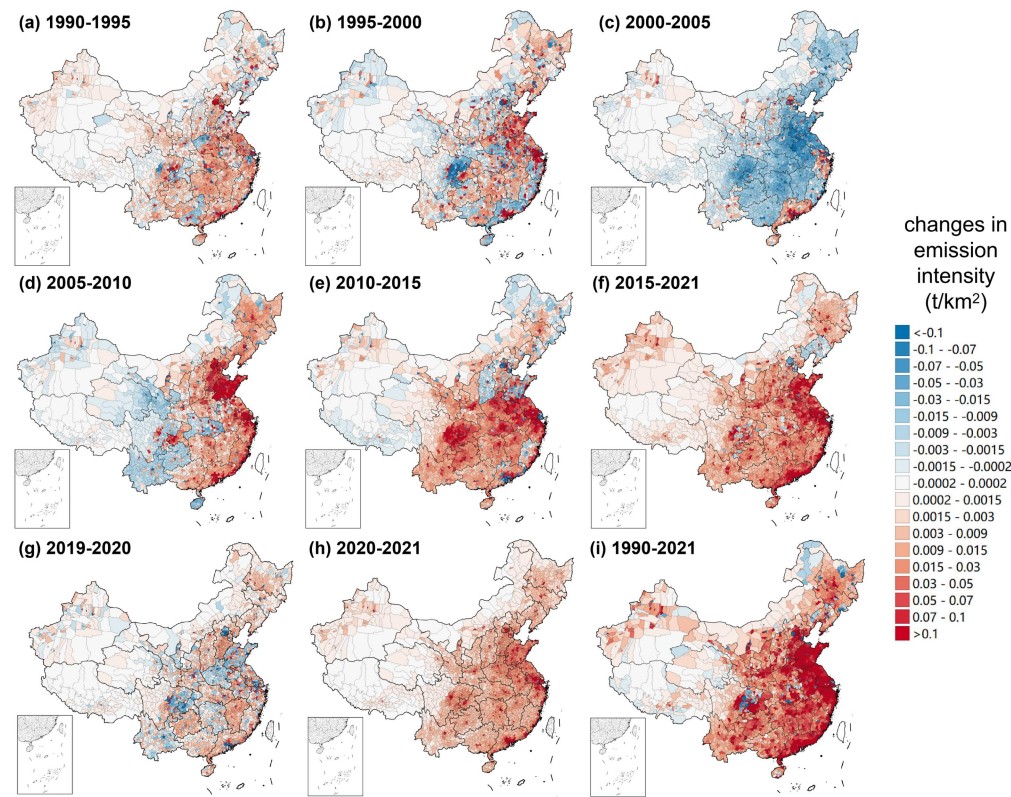

**Figure 6: Changes in organic emission intensity in each county during different periods.**

From 2000 to 2005, emissions declined in most parts of the country due to the implementation of China's first nationwide pollution control policy targeting cooking emissions in 2001 (see Section 3.3 for details) (State Environmental Protection Administration of China, 2001). However, Guangdong, Zhejiang, and Beijing maintained emission growth during this period, attributable to their exceptional economic expansion and continued population inflow (Kong, 2022; Zhu, 2012). From 2005 to 2010, cooking emissions in most counties in eastern China increased rapidly, likely because the emissions increase driven by rapid economic development outweighed the reductions from pollution control measures (Fleisher et al., 2010). In contrast, emissions in the slower-developing western regions decreased during this period (Fleisher et al., 2010).

During 2010-2021, emissions increased significantly nationwide except in provinces with stringent control policies (Beijing Environmental Protection Bureau, 2018; Feng et al., 2019; Liaoning Provincial Government, 2017; Shanxi Provincial Government, 2017). Coastal regions experienced the most notable emission increases, reflecting their faster economic growth and urbanization rates compared to inland regions (Shi, 2020). This rapid development also led to increased demand for cooking activities and a thriving catering industry due to population growth. Notably, Sichuan also exhibits significant emission increases, likely attributable to its special local cuisine that has attracted tourists nationwide and fostered a thriving catering industry (Li, 2017; Tian and Shen, 2024).

Additionally, we specifically examine the impact of the COVID-19 pandemic on cooking emissions from 2019 to 2021. In 2020, lockdown measures were implemented across China to control the spread of the pandemic (Chang et al., 2023). As shown in Fig. 6(g), cooking emissions in many regions decreased in 2020. For example, Beijing, the Yangtze River Delta, and the Pearl River Delta saw significant reductions in cooking emissions, likely because these areas originally had thriving catering industries that were heavily restricted by lockdown policies in 2020 (Lan et al., 2018; Li et al., 2021; Yuan et al., 2024), leading to a substantial decrease in commercial cooking emissions. Conversely, emissions increased in many other regions, likely because lockdown policies forced people to stay at home, shifting cooking and dining from centralized locations like canteens and restaurants to more dispersed cooking and dining at home (Yang et al., 2021), thereby increasing overall cooking emissions. In 2021, as lockdown policies were gradually relaxed and the catering industry began to recover, overall cooking emissions rebounded nationwide (Li et al., 2021).

The observations above indicate that our emission calculation methodology can effectively capture the influences of pivotal external factors affecting emissions. In the 1990s, changes in emissions across counties were primarily influenced by economic growth rates and population migration. After 2000, variations in emissions were likely influenced by the promotion of pollution control measures and the development of the catering industry. Overall, cooking emissions have increased in the vast majority of the country over the last three decades (1990–2021) as shown in Fig. 6(i), with particularly significant increases in the eastern region. Only a few counties have seen a reduction in emissions, typically coinciding with population changes.

## 4.2 National emission drivers and key influencing factors for county-level emissions

Based on sensitivity simulation, we find significant differences in the driving factors of China's cooking organic emissions during different periods. The decomposition of emission change drivers for each period is shown in Fig. 7. From 1990 to 2000, emission levels grew slowly, mainly driven by the increasing population and urbanization rate, which contributed 50.3% and 23.4% to the emission growth, respectively. From 2001 to 2005, while population growth and urbanization also promoted an increase in emissions, the implementation of emission standards in 2001 significantly strengthened pollution control measures (State Environmental Protection Administration of China, 2001), leading to a considerable reduction (-21%) in cooking emissions. From 2005 to 2015, while pollution emission standards continued to be enforced, the lack of new regulatory policies targeting cooking sources limited further pollution control effectiveness, resulting in an average annual reduction of only about 1% (Gao, 2020). Meanwhile, the rise in tertiary GDP and urbanization rates, marking rapid economic development, prompted a rapid increase in cooking emissions. Between 2005 and 2015, the rise in tertiary GDP and urbanization rates contributed 33.1% and 29.5% to the growth in emissions, respectively.

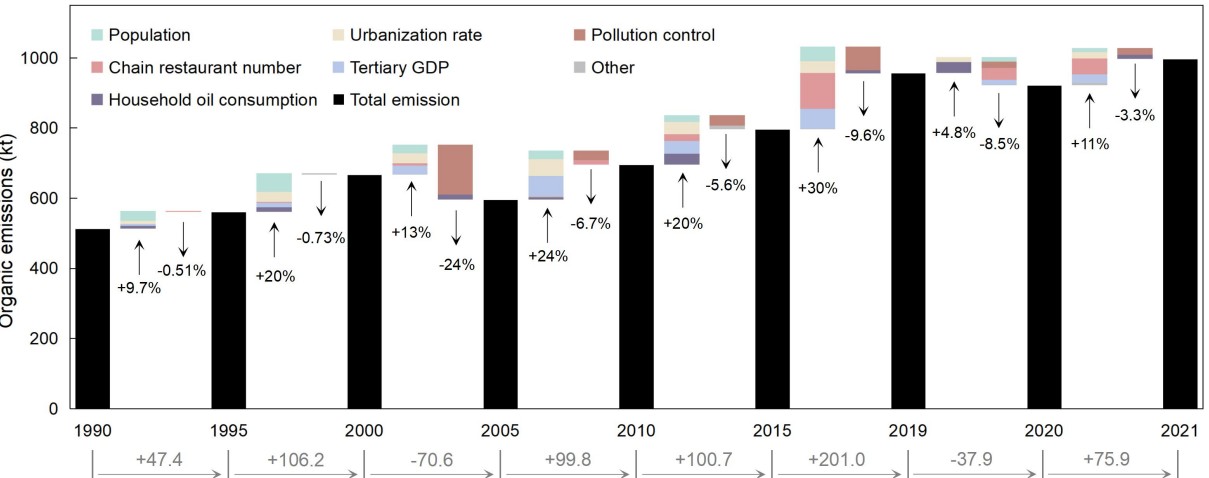

**Figure 7: The contribution of various driving factors to the changes in national cooking organic emissions across different periods.**

Since 2015, the increase in the number of chain restaurants has been the main driver for cooking emissions, possibly attributed to the prosperity of the catering industry brought about by online food delivery services (Maimaiti et al., 2018; Zhao et al., 2021). From 2015 to 2017, the number of users of online food delivery surged from 114 million to 343 million, and this figure continues to climb (Maimaiti et al., 2018). Besides, tertiary GDP, urbanization rate, and population also contribute to the growth of cooking emissions. Meanwhile, stricter pollution control measures led to a more pronounced reduction in emissions (achieving an 8.4% decrease from 2015 to 2019). However, this effect remained relatively limited compared to the rapid growth in total emissions (+30.0% during the same period). This suggests that existing regulations

were insufficient to address the growing emissions from the catering industry, highlighting the need for updated and more stringent policies specifically aimed at controlling cooking emissions. Overall, the primary driving factors of cooking organic emissions in the early (1990-2001), middle (2001-2015), and recent (2015-2021) periods are population growth, the rise in tertiary GDP and urbanization rates, and the increase in the number of chain restaurants.

We also explore the impact of the pandemic on the total national cooking emissions. From 2019 to 2020, factors such as the number of chain restaurants, population, and tertiary GDP were negatively affected by the pandemic, leading to a decrease in cooking emissions. However, some factors, including the urbanization rate and household cooking oil consumption, contributed to an increase in emissions. Despite the pandemic, China's urbanization rate rose from 60.6% in 2019 to 63.9% in 2020. This could be attributed to the Chinese government's efforts towards achieving the goal of *a moderately prosperous society in all respects before 2021* (Li, 2023), which involved continued urban development and infrastructure improvements. Additionally, the increase in household cooking oil consumption likely drove up emissions because lockdowns led to more people cooking at home rather than dining out (Yang et al., 2021). In 2021, as the economy recovers and the catering industry rebuilds, many factors (including the number of chain restaurants, population, and tertiary GDP) begin to lead the way again for increased cooking emissions (Li et al., 2021).

Additionally, we employ counterfactual analysis to quantitatively assess the emission reduction effects of control policy interventions. Specifically, we examine the two core policies introduced in Section 3.3: Policy-2001 and Policy-2013 (CPGPRC, 2013; State Environmental Protection Administration of China, 2001). While we collect provincial policies to determine province-specific PFIPs, these provincial regulations were largely driven by these two national policies, with variations in provincial response timing and enforcement stringency. Therefore, we sequentially exclude these two policies in our model, simulating counterfactual scenarios where provinces did not further strengthen pollution control measures after the policies were not implemented. The results are shown in Fig. S10. Without Policy-2013, 2021 emissions would have been 13.1% higher. Without both policies, the increase would reach 49.4%, highlighting the significant long-term impact of Policy-2001 in curbing emissions. The more modest effect of Policy-2013 may stem from its focus on comprehensive air pollution control, with catering sources being only a minor component. Moreover, the 2013 policy and subsequent provincial policies primarily emphasized increasing the installation rates of catering fume purification devices without stringent requirements for ensuring the removal efficiency of installed equipment (CPGPRC, 2013). Due to equipment aging and inadequate supervision (Li et al., 2023), the average removal efficiency remained low, resulting in no significant emission reductions despite higher installation rates.

Furthermore, we also pay attention to the key influential factors of cooing emissions of various counties at different development stages, applying the SHAP algorithm for the quantitative analysis. Fig. S11 presents an overview of the SHAP values for each factor influencing emissions of the three cooking emission sectors, with the y-axis sorted from high to low based on the impact of each factor on emissions. The influencing factors of commercial cooking emissions are the most complex. Urbanization rate (UR), population (POP), and EFs are the top three factors that have the greatest impact on

commercial cooking emissions of counties, with increasing values leading to emissions growth. Additionally, PFIP, the tertiary GDP (GDP3), per capita household edible oil consumption (HOC), the number of chain restaurants (NCR), and per capita disposable income (DI) all affect commercial cooking emissions to some extent. Additionally, residential cooking
emissions are mainly influenced by population and per capita household edible oil consumption. Canteen cooking emissions are mainly affected by population, PFIP, and the population of employees in enterprises (PEE).

We further analyze the marginal effects of each influencing factor on the cooking organic emissions, that is, how emission values (indicated by SHAP values) vary with the values of individual influencing factors. Taking commercial cooking emissions as an example, the partial dependence plot of SHAP values on the main influencing factors is shown in Fig. 8. For
the urbanization rate, the relationship between SHAP values and the urbanization rate forms an S-shaped curve. This means that the sensitivity of commercial cooking emissions to the urbanization rate is relatively high when the urbanization rate is at the medium level (45%-75%). Additionally, the SHAP values are approximately linearly correlated with the local population and EFs, while the emissions are negatively correlated with the PFIP value. The relationship between the tertiary GDP and the number of chain restaurants and SHAP values approximates a logarithmic growth curve, where growth is rapid
at lower feature values and slows down as the feature values increase. The relationship between HOC and commercial cooking emissions is very intricate. When the HOC value is low, its increase signifies an improvement in people's living standards starting from a low level, which in turn leads to a corresponding increase in commercial cooking emissions. As the HOC value reaches a certain level, further increases indicate an increase in the frequency of residential cooking that competes with commercial cooking, resulting in a decrease in commercial cooking emissions. Finally, an overall increase in
630   per capita disposable income will lead to an increase in commercial cooking emissions, as this can be explained by people having more funds for dining out. The relationship between residential cooking emissions and its main influencing factors (population and HOC), as well as the relationship between canteen cooking emissions and its main influencing factors (POP and PFIP), is very similar to the relationship between commercial cooking emissions and these variables.

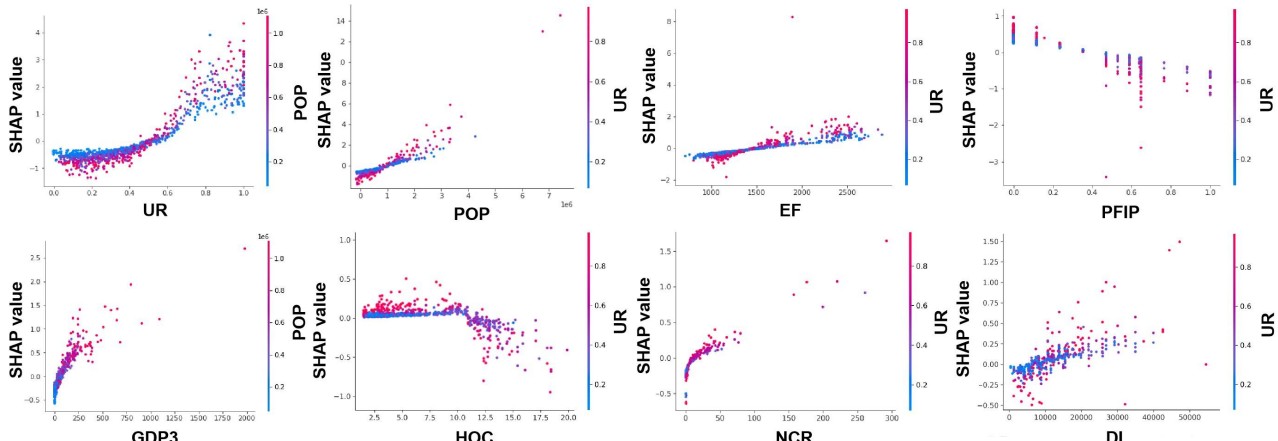

 **Figure 8: The partial dependence plot of SHAP values on the main influencing factors of commercial cooking organic emissions in Chinese counties.**

## 5 Data availability

The county-level cooking emission inventory in China from 1990 to 2021 is publicly available at the repository (https://doi.org/10.6084/m9.figshare.26085487) (Li et al, 2025). This dataset provides comprehensive emissions data at the county level, covering all 2,848 counties in mainland China based on the 2020 administrative divisions, and includes annual emissions for every year from 1990 to 2021. The emissions are categorized by subsectors, including commercial cooking, residential cooking, and canteen cooking, and by pollutants, including organics across the full volatility range (VOCs, SVOCs, IVOCs, and xLVOCs), $PM_{2.5}$, UFPs, and PAHs. The types of emission pollutants related to PAHs include gaseous PAHs, particulate PAHs, and $BaP_{eq}$. Besides, the main text and supplementary materials (Table S2-S5) also provide detailed listings of emission factors, PFIPs, PAHs' TEF, and other parameters used for calculating emissions. Additionally, the input data for the machine learning models, such as population, economic, and catering-related statistical indicators, are sourced from the Chinese County Statistical Yearbook, China Urban Statistical Yearbook, and China Market Statistics Yearbook, with a full description provided in Table S1 (National Bureau of Statistics of China, 2022a, c, b).

## 6 Conclusions and implication

In this study, leveraging machine learning to overcome the challenges of obtaining activity data, we establish China's first county-level cooking emission inventory, with a temporal scale extending back to 1990. Unlike previous inventories that relied on proxy data such as population for calculation and downscaling, our inventory employs a powerful ensemble machine learning model to capture the complex relationships between county-level cooking activities and factors involving population, economics, and the catering industry. This method enables direct calculation of emissions at the county level, resulting in spatial distributions that better reflect real-world conditions. Moreover, our method can sensitively identify the impact of external factors, such as the COVID-19 pandemic and the rise of food delivery services, on cooking emissions. Based on this accurate, high-resolution, and long-term inventory, we have updated the scientific understanding of the spatiotemporal trends and driving forces of cooking emissions.

Given that cooking is a significant source of $PM_{2.5}$ (Yuan et al., 2023), our long-term, high-spatial-resolution cooking emission inventory provides essential data for accurately simulating $PM_{2.5}$ concentrations and conducting precise source apportionments at large spatiotemporal scale. Furthermore, for the first time, we incorporate UFPs and PAHs into the national cooking emission inventory, filling a gap in studies on the health impact of cooking emissions. Previous studies on the health impacts of cooking emissions primarily focused on indoor environments (Chen et al., 2018; Zhang et al., 2023; Zhao and Zhao, 2018). However, pollutants emitted into the outdoor atmosphere from cooking may also have significant

health risks, due to the proximity of cooking emission sources to the human living environment. Our accurate, high-resolution cooking inventory, combined with the inclusion of highly toxic pollutants, provides critical but previously missing data for assessing exposure risks to cooking-related pollutants in outdoor environments. This enables a comprehensive understanding of the health impacts of cooking emissions by integrating both indoor and outdoor exposure assessments.

Our identification of the spatiotemporal patterns and driving factors of national cooking emissions also provides valuable insights for targeted policy formulation. With the significant reduction in emissions from sectors such as industry and energy, the critical impact of cooking emissions is becoming increasingly prominent and may become a major source in the future (Zhao and Zhao, 2018). However, our results indicate that existing control measures are insufficient to curb the rapid growth of cooking emissions, necessitating the development of updated and more effective control strategies. Given that cooking is a fundamental human need, reducing emissions by restricting cooking activities or altering dietary habits is not feasible. A more viable approach is to enhance end-of-pipe treatment. However, cooking emission sources are numerous and widespread, making comprehensive control efforts highly labor-intensive. Fortunately, our high-resolution inventory reveals a strong spatial coupling between high emission intensity and population density. Specifically, 30% of counties—covering only 14.5% of the national land area—contribute over 60% of cooking-related organic emissions while housing 60% of the population. These areas face substantial population exposure risks, and prioritizing stricter controls there could effectively mitigate health impacts. Additionally, the limited effectiveness of recent pollution control policies may stem from their sole focus on the installation rate of purification facilities while neglecting actual purification efficiency. Therefore, future efforts should establish stricter regulations or standards for purification efficiency and strengthen enforcement. Besides, residential emissions have consistently accounted for ~30% of total cooking emissions, yet targeted policies and specialized purification facilities remain scarce. Potential solutions include developing compact purifiers for household kitchens and implementing exhaust purification systems for residential chimneys. Finally, our fine-scale emission inventory can serve as a key input for the air quality modeling and control strategy optimization model, enabling further exploration of differentiated mitigation strategies that consider costs, health benefits, and local resource capacity.

Additionally, the methodology adopted in this study also offers a reference for the long-term and accurate estimation of emissions from other sources and other regions. We innovatively use counties as the basic unit to estimate emissions, which not only provides the machine learning model with rich and wide-span county samples at different development stages, enhancing the model's performance, but also ensuring a high spatial resolution. Besides, the data used for machine learning modeling are also readily available, significantly reducing the difficulty of activity level acquisition. Similar to cooking emissions, emissions from domestic combustion, for example, can be estimated using statistical indicators such as temperature, per capita disposable income, urbanization rate, and energy consumption. In other regions, this methodology also shows potential in estimating high-resolution emissions through machine learning models and localized datasets. This contributes to more comprehensive and accurate research on air pollution.

We also acknowledge some limitations of our study. For county-level emissions, while SHAP provides interpretable insights into feature importance, it cannot infer causality to identify the underlying driving factors. Future work could employ causal inference techniques such as counterfactual prediction and difference-in-differences analysis (Dong et al., 2022; Li et al., 2024a) to more accurately assess the actual impacts of policy interventions and socioeconomic factors on emission trends, thereby providing more robust evidence to support localized emission reduction policies.

## Author contributions

ZL, SW, BZ, and SL designed the study. ZL developed the emission inventory. SL, ZS, XY, and GH provide key data for the calculation of the emission inventory. DY, QW, FZ, XY, and YZ helped to improve the emission inventory. ZL wrote the original draft; all the coauthors revised the manuscript.

## Competing interests

The authors declare that they have no conflict of interest.

## Disclaimer

Publisher's note: Copernicus Publications remains neutral with regard to jurisdictional claims in published maps and institutional affiliations.

## Financial support

This work is supported by National Key Research and Development (R&D) Program of China (2022YFC3702905), National Natural Science Foundation of China (grant 22188102) and the Samsung Advanced Institute of Technology.

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
