# Peer review of "Tracking County-level Cooking Emissions and Their Drivers in China from 1990 to 2021 by Ensemble Machine Learning"

_Earth System Science Data, 2025_

## Author Comment (AC1)

**Dear editor and the anonymous reviewers,**

Thanks a lot for your work and time on our manuscript.

The paper entitled "*Tracking County-level Cooking Emissions and Their Drivers in China from 1990* to 2021 by Ensemble Machine Learning" (Manuscript ID: essd-2025-104) by Zeqi Li, et al., has been revised carefully according to the correction requests and review reports.

The authors have addressed all the reviews' comments point-by-point as below. All the corrections and responses have been incorporated into the revised manuscript and supplement (marked with **BLUE COLORED FONTS**).

If further responses and corrections should be made, please don't hesitate to let us know.

Sincerely, Corresponding author Prof. Shuxiao Wang School of Environment, Tsinghua University Beijing 100084, P.R. China E-mail: shxwang@tsinghua.edu.cn

**Reply on RC1:**

**Dear reviewer,**

Thank you very much for your recognition and the valuable suggestions! We have addressed the comments point-by-point as below.

All the corrections have been incorporated into the revised manuscript (manuscript\_R1) and the revised supplement (supplement\_R1). The point-to-point responses are listed as follows. If further responses and corrections should be made, please don't hesitate to let us know.

**Comment 1:**

Introduction: Consider briefly introducing unique characteristics of Chinese cooking and the special requirements of these characteristics for the construction of emission inventories, which will help international readers better understand the importance of the research.

**Response 1:**

Thank you for the insightful comments and suggestions to help enhance the international appeal of our manuscript. Based on your suggestions, we have supplemented the relevant information about Chinese cooking in the introduction (lines 59-66):

Chinese cooking emissions exhibit unique characteristics that require special consideration in emission inventory development (Zhao and Zhao, 2018). The widespread use of high-temperature oils, various seasonings, and special techniques like stir-frying generates pollutants with complex chemical compositions (Chen et al., 2018; Zhao and Zhao, 2018). Furthermore, cooking styles vary greatly across different regions in China, which have historically developed into multiple distinct cuisine systems (Li et al., 2023; Lin et al., 2022b). To accurately capture these emission patterns and quantify their impact, the emission estimate needs to incorporate cuisine-specific multi-pollutant emission factors while explicitly accounting for regional variations through spatial resolution representation (Lin et al., 2022b; Zhao and Zhao, 2018).

We believe this addition will help international readers better understand the unique characteristics and importance of Chinese cooking.

**Comment 2:**

Line 165-166: Clarify how "variables of lower importance" were determined (e.g., specific threshold for RF feature importance scores).

**Response 2:**

Thank you for your valuable suggestion. We have added the following sentence (lines 186-189) of the manuscript\_R1 to clarify how we determined "variables of lower importance":

Variables were first ranked by RF model importance scores, with those scoring <5% considered for potential removal because of low importance (Alduailij et al., 2022; Ye et al., 2022). Each candidate variable was then temporarily excluded, and only those whose removal resulted in an R2 decline of <1% were permanently discarded – ensuring no critical features were lost (Altmann et al., 2010; Zhu et al., 2023).

**Comment 3:**

Line 215-217: "directly calculates county-level cooking emissions" is inaccurate. The emissions of this study are still estimated through machine learning predictions, not direct estimates, so the statement needs to be revised.

**Response 3:**

Thank you for your insightful comment. We agree that the term "directly calculates" could be inaccurate. To better describe our methodology, we have revised the wording to "precisely estimates", which more accurately reflects the advantage of our approach - by conducting activity-level modeling and emission estimation at the county-level spatial scale from the outset, we ensure high-resolution results. The revised sentence (lines 25-27) now reads:

Unlike previous inventories that rely on simplistic proxy data such as population for calculation and downscaling, our inventory precisely calculates county-level cooking emissions, providing more accurate emission estimates and spatial distributions.

**Comment 4:**

Figure 3: Provide sector-specific spatial distributions (commercial/residential/canteen) for a representative year in the supplement.

**Response 4:**

Thank you for your suggestion. We have added a figure (Figure S6) in the supplement\_R1 showing the sector-specific spatial distributions of cooking emissions for the representative year (2021):

Figure. S6. Sector-specific spatial distributions of cooking emissions in 2021.

Additionally, we have added the following sentence (lines 376-377) in the main text to guide readers to this figure:

Fig. 3 provides high-resolution spatial distribution maps of cooking organic emissions in China from 1990 to 2021, while Fig. S6 provides the sector-specific spatial distributions of cooking organic emissions for a representative year (2021).

**Comment 5:**

Lines 332-333: Provide percentage contributions of key regions (Beijing-Tianjin-Hebei, Yangtze Delta, etc.) to national total emissions.

**Response 5:**

Thank you for your suggestion. We have supplemented the percentage contributions of key regions to the national total emissions in the revised manuscript (lines 384-386). The updated text now reads:

The North China Plain region, the Yangtze River Delta, the Pearl River Delta, and the Sichuan-Chongqin region became the four key emission zones, contributing 20.2%, 19.9%, 8.63%, and 7.98%, respectively, of the nation's total cooking organic emissions.

**Reference:**

- Alduailij, M., Khan, Q. W., Tahir, M., Sardaraz, M., Alduailij, M., and Malik, F.: Machine-Learning-Based DDoS Attack Detection Using Mutual Information and Random Forest Feature Importance Method, Symmetry, 14, 1095, https://doi.org/10.3390/sym14061095, 2022.
- Altmann, A., Toloşi, L., Sander, O., and Lengauer, T.: Permutation importance: a corrected feature importance measure, Bioinformatics, 26, 1340–1347, https://doi.org/10.1093/bioinformatics/btq134, 2010.
- Chen, C., Zhao, Y., and Zhao, B.: Emission Rates of Multiple Air Pollutants Generated from Chinese Residential Cooking, Environ. Sci. Technol., 52, 1081–1087, https://doi.org/10.1021/acs.est.7b05600, 2018.
- Li, Z., Wang, S., Li, S., Wang, X., Huang, G., Chang, X., Huang, L., Liang, C., Zhu, Y., Zheng, H., Song, Q., Wu, Q., Zhang, F., and Zhao, B.: High-resolution emission inventory of full-volatility organic compounds from cooking in China during 2015–2021, Earth System Science Data, 15, 5017–5037, https://doi.org/10.5194/essd-15-5017-2023, 2023.
- Lin, P., Gao, J., Xu, Y., Schauer, J. J., Wang, J., He, W., and Nie, L.: Enhanced commercial cooking inventories from the city scale through normalized emission factor dataset and big data, Environmental Pollution, 315, 120320, https://doi.org/10.1016/j.envpol.2022.120320, 2022.
- Ye, X., Wang, X., and Zhang, L.: Diagnosing the Model Bias in Simulating Daily Surface Ozone

Variability Using a Machine Learning Method: The Effects of Dry Deposition and Cloud Optical Depth, Environ. Sci. Technol., 56, 16665–16675, https://doi.org/10.1021/acs.est.2c05712, 2022.

- Zhao, Y. and Zhao, B.: Emissions of air pollutants from Chinese cooking: A literature review, Build. Simul., 11, 977–995, https://doi.org/10.1007/s12273-018-0456-6, 2018.
- Zhu, J.-J., Yang, M., and Ren, Z. J.: Machine Learning in Environmental Research: Common Pitfalls and Best Practices, Environ. Sci. Technol., 57, 17671–17689, https://doi.org/10.1021/acs.est.3c00026, 2023.

---

## Author Comment (AC2)

**Dear editor and the anonymous reviewers,**

Thanks a lot for your work and time on our manuscript.

The paper entitled "*Tracking County-level Cooking Emissions and Their Drivers in China from 1990 to 2021 by Ensemble Machine Learning*" (Manuscript ID: essd-2025-104) by Zeqi Li, et al., has been revised carefully according to the correction requests and review reports.

The authors have addressed all the reviews' comments point-by-point as below. All the corrections and responses have been incorporated into the revised manuscript and supplement (marked with **BLUE COLORED FONTS**).

If further responses and corrections should be made, please don't hesitate to let us know.

Sincerely, Corresponding author Prof. Shuxiao Wang School of Environment, Tsinghua University Beijing 100084, P.R. China E-mail: shxwang@tsinghua.edu.cn

**Reply on RC2:**

Dear reviewer,

Thank you very much for your recognition and the valuable suggestions! We have addressed the comments point-by-point as below.

All the corrections have been incorporated into the revised manuscript (manuscript\_R1) and the revised supplement (supplement\_R1). The point-to-point responses are listed as follows. If further responses and corrections should be made, please don't hesitate to let us know.

**Comment 1:**

In the Introduction, it is recommended to more systematically review existing domestic and international methodologies for constructing cooking emission inventories, along with their limitations. The study's innovations in spatiotemporal resolution, pollutant coverage, and emission estimation accuracy should be quantitatively emphasized. In addition, the discussion of health risks associated with cooking emissions would benefit from stronger empirical evidence. The roles of ensemble machine learning and SHAP in addressing key scientific challenges should also be clarified, thereby strengthening the logical connection between methodological choices and research objectives.

**Response 1:**

We sincerely appreciate the valuable suggestions from the reviewers, which have significantly improved our Introduction section. We have made the following additions:

(1) We have systematically revised and supplemented the review of domestic and international methodologies for constructing cooking emission inventories and their limitations (lines 67-99):

Internationally, some efforts have been made to develop cooking emission inventories. High-resolution emission datasets have been established for small-scale regions, such as Greater Athens in Greece and the Red River Delta in Vietnam, through field surveys and measurements (Fameli et al., 2022; Huy et al., 2021). However, at larger scales (e.g., national or global), cooking sources are often omitted from anthropogenic emission inventories or only roughly estimated using uniform emission factors and simplistic statistics like food supply or meat consumption (Huang et al., 2023; Saha et al., 2024). These methods and data are difficult to apply to China because, as mentioned above, cooking inventories in China require localized emission factors and estimation methods that explicitly consider regional differences.

Domestic inventories also exhibit the characteristic of being "precise at small scales but coarse at large scales," making it difficult to balance accuracy and breadth (Cheng et al., 2022; Jin et al., 2021; Liang et al., 2022; Wang et al., 2018a). These limitations are mainly due to the difficulty in obtaining high-quality data, particularly activity level data, over large spatiotemporal scales and at fine spatial resolutions. Some studies have collected key data for emission calculations by cuisine-specific emission factor testing, door-to-door surveys of restaurants and online fume monitoring systems, and thereby established high-resolution inventories of single years in cities or districts such as Beijing, Shanghai, and Shunde (Lin et al., 2022b; Wang et al., 2018b, 2018a; Yuan et al., 2023). These studies have provided valuable localized basic data

for China's cooking emission inventories. However, obtaining accurate cooking activity data (e.g., restaurant numbers) remains challenging at larger temporal and spatial scales. Traditional China's national cooking emission inventories either use simplistic statistical data (such as population and catering consumption expenditure) as proxies for activity levels, or linearly extrapolate the activity levels of one city to other areas based on these simple statistics (Cheng et al., 2022; Jin et al., 2021; Liang et al., 2022; Wang et al., 2018a). These simplifications and linear assumptions result in high uncertainties and low spatial resolution. Recent studies have more accurately estimated national cooking emissions based on data from digital maps or catering service platforms (Li et al., 2023; Zhang et al., 2024b). However, these inventories are limited to recent years, as they rely on newly developed data platforms.

Apart from lacking accuracy and breadth, another limitation of existing cooking emission inventories is their limited pollutant coverage. Previous studies on cooking emissions primarily focused on PM2.5 (whose organic component is primary organic aerosol, POA) and volatile organic compounds (VOCs) (Jin et al., 2021; Wang et al., 2018a, 2018b). However, recent advancements in the framework for organic compounds in the full volatility range (including VOCs, intermediate-volatility organic compounds (IVOCs), semi-volatile organic compounds (SVOCs), and organic compounds with even lower volatility (xLVOCs)) have revealed the previously overlooked significant contributions of I/SVOCs to secondary organic aerosols (SOAs) (Chang et al., 2022; Zhang et al., 2021). Although our latest study has supplemented the inventory with organics in the full volatility range (Li et al., 2023), the emissions for certain highly toxic pollutants of particular concern emitted from cooking- notably ultrafine particles (UFPs) and polycyclic aromatic hydrocarbons (PAHs) - remain lacking (Chen and Zhao, 2024; Jørgensen et al., 2013; Lachowicz et al., 2023; Lin et al., 2022a). This gap limits our comprehensive assessment of the environmental and health risks associated with cooking emissions.

We have also added a summary of the limitations of existing inventories (lines 115-118):

In summary, limited by the difficulty in obtaining high-quality activity data, there is currently a lack of an accurate, long-term, high-resolution national cooking emission, and existing inventories remain deficient in their coverage of important toxic pollutants such as PAHs and UFPs. This hinders studies on PM2.5 modeling, source apportionment, and health risk analysis.

(2) We have quantitatively emphasized the innovations of this study in spatiotemporal resolution, pollutant coverage, and emission estimation accuracy (lines 130-138).:

We expect to achieve breakthroughs in spatiotemporal resolution, pollutant coverage, and emission estimation accuracy. This study represents the first long-term (nearly 31 years) high-resolution (county-level) inventory, whereas existing national inventories were mostly limited to single years or recent years at provincial resolution. Besides, our study covers key pollutant categories from cooking emissions, including organics in the full volatility range, PAHs, and UFPs that were not included in other national inventories. In terms of estimation accuracy, we adopted cuisine-specific emission factors, considered dynamically changing purification facility installation proportions (PFIPs) driven by provincial policies, and used precise county-level activity data to calculate emissions more accurately and better reflect regional differences. Finally, this study will provide important data and new perspectives for researching the impacts of cooking emissions on air pollution and human health, facilitating the development of targeted emission control policies.

(3) Regarding the health risks associated with cooking emissions, as recommended, we have supplemented stronger empirical evidence, including specific hazardous components, disease association evidence, toxicity experimental results, and exposure risk assessment studies. Specifically, we have added the following content to the main text (lines 49-55).

Moreover, cooking emissions contain multiple hazardous components, such as ultrafine particles (UFPs) and polycyclic aromatic hydrocarbons (PAHs), which are linked to health problems including cardiovascular disease, oxidative stress, and lung cancer (Kim et al., 2024; Lin et al., 2022b; Naseri et al., 2024; Xu et al., 2020). Experiments have proved that both gaseous organics and PM2.5 emitted from cooking exhibit much more negative biological effects like cytotoxicity compared to ambient PM2.5 (Guo et al., 2023). Consequently, cooking emissions can increase PM2.5 concentrations and toxicity, thereby exacerbating air pollution and associated disease burdens (Chafe et al., 2014; Wang et al., 2017; Zhang et al., 2024a).

(4) Regarding ensemble machine learning and SHAP, we have supplemented their roles in addressing key scientific challenges as recommended. First, we introduce the roles of ensemble machine learning and SHAP in addressing key scientific challenges in atmospheric science. The updated content appears in lines 100-110 of the main text:

In recent years, machine learning has been widely applied in atmospheric pollution research due to its powerful capability to process large-scale spatiotemporal datasets and capture complex nonlinear relationships within them (Liu et al., 2023; Prodhan et al., 2022a; Zhang and Zhao, 2024; Zheng et al., 2021). Models such as Random Forest (RF), eXtreme Gradient Boosting (XGBoost), and Deep Neural Networks (DNN) have demonstrated strong performance in predicting pollutant concentration time series and identifying spatial distributions (Chen et al., 2024; Prodhan et al., 2022b; Ren et al., 2022; Wu et al., 2024; Xu et al., 2023). Ensemble machine learning models further achieve better and more stable results by combining predictions from individual base models (Liu et al., 2023; Ren et al., 2022). They can help supplement sparse datasets, serving as an effective alternative for obtaining key data that would otherwise be computationally expensive or inaccessible to collect (Ren et al., 2022; Shi et al., 2024; Xiao et al., 2018). When integrating machine learning with SHapley Additive exPlanations (SHAP) additivity algorithm, the key factors for the predictive target and their influence patterns can be identified (Hou et al., 2022; Yang et al., 2023).

Meanwhile, we also explain how these models help us overcome key challenges in our study. The additional content appears in lines 110-114 of the main text:

Most importantly, these approaches hold significant potential to address key challenges in cooking emission inventory development. Where conventional activity data at large spatiotemporal scales are unavailable, ensemble machine learning models can predict long-term, high-resolution activity levels by capturing complex relationships between cooking activities and fundamental socioeconomic indicators. Coupled with SHAP analysis, they can provide insights into how socioeconomic factors influence emission trends. However, such efforts have not yet been made.

**Comment 2:**

In Section 2.1, the high-resolution cooking activity data used for model training only cover the period from 2015 to 2021. It remains unclear whether these data are sufficient to support reliable backcasting of activity levels for earlier periods such as the 1990s and early 2000s. The authors are advised to explicitly discuss the potential temporal bias introduced by this limited

training window and to clarify whether any sensitivity analysis or validation was conducted to assess its impact on historical emission estimates and associated uncertainties.

**Response 2:**

Thank you for your comment. We acknowledge that the limited training window may introduce potential temporal bias to the historical emission estimates. However, since earlier-period data are unavailable, we have employed models with the best possible extrapolation capabilities for estimation, which have demonstrated significant improvements over traditional methods. Following your suggestion, we have supplemented the manuscript with: (1) explicit acknowledgment of potential bias, (2) sensitivity analysis validating the model's extrapolation capability through stratified testing across counties at different development stages, (3) cross-validation using trends from other relevant statistical data, and (4) discussion of how this bias affects uncertainties in historical emission estimates. We believe these additions enhance the scientific rigor of our study. The specific revisions are as follows:

In the methods section (2.1), we added a note (lines 206-208) about potential bias introduced by modeling with 2015–2021 data, emphasizing subsequent validation and uncertainty evaluation:

Notably, the modeling data only covers the period from 2015 to 2021, as earlier data are unavailable. This may introduce some bias when backcasting activity levels for earlier periods. We further validate the backcasted activity levels and evaluate their uncertainties.

In the model evaluation section (3.1), we added a discussion (lines 354-367) on temporal bias from the limited training window, including sensitivity analysis and validation:

While the model demonstrates good performance for near-term extrapolation, greater uncertainty may exist when backcasting to earlier periods, which include more underdeveloped counties. To evaluate this, we conduct sensitivity analyses by training on the top 70% GDP-ranked counties and testing on the bottom 30%. For commercial catering (the most complex case, as shown in Fig. S4), the ensemble models test-set R2 remained robust (R2=0.719), outperforming the best statistical models (R2=0.523). For real-world backcasting of historical data, the 2015 - 2021 training data already include some less-developed regions that can represent early-stage conditions, mitigating extreme extrapolation risks. Additionally, we further validate the historical trends of predicted activity data based on the limited available historical data (Fig. S5). From 1990 to 2021, the growth rate of commercial cooking activity levels was intermediate between population growth and tertiary GDP. The temporal evolution resembles that of the chain restaurant number (slow early growth followed by acceleration), though the chain restaurant number is more stable as they exclude small independent restaurants. We also incorporate chain restaurant revenue data (available since 2004), which corroborates the fluctuations in our predictions including the post-2015 rapid growth driven by food delivery platforms and the 2020 pandemic-driven decline (Maimaiti et al., 2018; Zhao et al., 2021). Therefore, while temporal extrapolation may introduce biases (uncertainties are quantified in Section 3.3), our multi-pronged validation demonstrates the reasonableness and optimality of our backcasted estimates.

Figure S4. Sensitivity test of the best statistical model (power function regression) and ensemble machine learning model in extrapolating activity levels for counties with different GDP levels.

Figure S5. Comparison of the historical trends of predicted national commercial cooking activity levels with other relevant statistical data. For comparability, all data are normalized to the ratio relative to their 2021 values.

In the emission results (Section 3.3), we have considered the impact of backcasting biases on historical emission estimates and re-estimated the uncertainty ranges of emissions. The revised results have been updated in the main text, and the uncertainty estimation methods have been detailed in the supplement. In the lines 420-422 in the main text, we revised:

The uncertainty ranges are determined through Monte Carlo simulations referencing previous studies (Chang et al., 2022;

**Nan Li, 2017), incorporating cumulative biases introduced by extrapolating historical emissions using limited training data (see Text S3 for details).**

The revised uncertainty ranges have been updated in Figure 4: